# Radiocaesium accumulation capacity of epiphytic lichens and adjacent barks collected at the perimeter boundary site of the Fukushima Dai-ichi Nuclear Power Station

Terumi Dohi[1]*, Yoshihito Ohmura[2], Kazuya Yoshimura[3], Takayuki Sasaki[4], Kenso Fujiwara[1], Seiichi Kanaizuka[5], Shigeo Nakama[3], Kazuki Iijima[1]

1 Sector of Fukushima Research and Development, Japan Atomic Energy Agency, MiharuTown, Tamura-gun, Fukushima, Japan, 2 Department of Botany, National Museum of Nature and Science, Tsukuba-City, Ibaraki, Japan, 3 Sector of Fukushima Research and Development, Japan Atomic Energy Agency, Minamisoma-city, Fukushima, Japan, 4 Department of Nuclear Engineering, Kyoto University, Kyoto-city, Kyoto, Japan, 5 Nuclear Engineering Co., Ltd., Tokai-town, Ibaraki, Japan

* dohi.terumi@jaea.go.jp

## Abstract

We investigated the radiocaesium content of nine epiphytic foliose lichens species and the adjacent barks of *Zelkova serrata* (Ulmaceae, "Japanese elm") and *Cerasus* sp. (Rosaceae, "Cherry tree") at the boundary of the Fukushima Dai-ichi Nuclear Power Station six years after the accident in 2011. Caesium-137 activities per unit area (the $^{137}$Cs-inventory) were determined to compare radiocaesium retentions of lichens (65 specimens) and barks (44 specimens) under the same growth conditions. The $^{137}$Cs-inventory of lichens collected from *Zelkova serrata* and *Cerasus* sp. were respectively 7.9- and 3.8-times greater than the adjacent barks. Furthermore, we examined the radiocaesium distribution within these samples using autoradiography and on the surfaces with an electron probe micro analyzer (EPMA). Autoradiographic results showed strong local spotting and heterogeneous distributions of radioactivity in both the lichen and bark samples, although the intensities were lower in the barks. The electron microscopy analysis demonstrated that particulates with similar sizes and compositions were distributed on the surfaces of the samples. We therefore concluded that the lichens and barks could capture fine particles, including radiocaesium particles. In addition, radioactivity was distributed more towards the inwards of the lichen samples than the peripheries. This suggests that lichen can retain $^{137}$Cs that is chemically immobilised in particulates intracellularly, unlike bark.

## Introduction

Lichens are symbiotic organism composed of fungi and algae that are well known to incorporate and store a variety of radionuclides derived from both natural origins, like U-238, Ra-226, Th-232 etc. in mine areas [1, 2], and artificial origins such as atmospheric nuclear weapon tests

**Data Availability Statement:** All relevant data are within the paper and its Supporting Information files.

**Funding:** This work was financially supported by an internal budget of Japan Atomic Energy Agency for the Fukushima reconstruction research activity. This research was supported by the MEXT Nuclear Energy S&T and Human Resource Development Project through Concentrating Wisdom Grant Number JPMX15D15664655 operated by KI, and Grant-in-Aid for Challenging Exploratory Research of JSPS (no.16K12627) operated by TD. These funders did not play a role in the study design, data collection and analysis, decision to publish, or preparation of the manuscript. They provided only financial support in the research materials in this study. Nuclear Engineering Co., Ltd. employed SK, this commercial affiliation did not play a role in this study.

**Competing interests:** The authors have declared that no competing interests exist. Nuclear Engineering Co., Ltd. which is author's affiliation, does not alter our adherence to PLOS ONE policies on sharing data and materials.

and nuclear power plant accidents [3–7]. Lichens are recognized as useful biomonitoring tools because of having the following features: (i) they are widely distributed in terrestrial ecosystems and found on various substrata such as rock, tree, soil, artificial man-made structures, and so on, (ii) they can accumulate metals including radionuclides directly absorbing from air into the body, due to their lack of root system and protective outer cuticle, (iii) they have high accumulation capacities of pollutants on the large surface areas of thalli, and (iv) they have longevity due to their slow metabolic activity and slow growth rate throughout the year [8–10].

Among the radionuclides released from the 1986 Chernobyl Nuclear Power Plant (CNPP) accident in USSR (26 April 1986), $^{137}$Cs has a long half-life of 30 years (compared to 2 years for $^{134}$Cs) and was transported over 9,000 km from the CNPP [11]. $^{137}$Cs has been studied extensively with respect to the food chain and human exposure effects, as it migrates via water, soil, air and plants similarly to potassium [12]. Thus, even 30 years after the CNPP accident, lichens were widely used in the biomonitoring researches for spatial and temporal deposition patterns of radionuclides, especially radiocaesium (e.g. [13–19]).

After the Fukushima Dai-ichi Nuclear Power Plant (FDNPP) accident resulting from the massive earthquake and tsunami on 11 March 2011, several radionuclides were released such as $^{133}$Xe, $^{131}$I, $^{132}$Te and $^{134, 136, 137}$Cs [11, 20, 21]. During the accident progression, $^{129, 129m}$Te, $^{134, 136, 137}$Cs and $^{110m}$Ag were detected at fallout monitoring stations in Fukushima Prefecture. Some of the highest recorded activities were for $^{134, 137}$Cs (140,000–5,600,000 MBq/km$^2$ month), between 1 March and 8 April 2011 [22]. A number of studies have confirmed that lichens are useful biomonitoring tools within areas affected by the FDNPP accident, i.e. $^{137}$Cs activity concentrations in foliose lichens reflected fallout levels, and by initial $^{137}$Cs activity concentrations in lichens could be evaluated retrospectively based on the temporal variation of activity concentrations [23–27].

In addition to lichens and mosses, tree barks were also studied as a biomonitoring tool for radionuclides, due to accumulation of radionuclides on bark surfaces [28–30]. While the accumulation (incorporation and retention) of radionuclides in lichens has been studied in many researches (e.g. [19, 31–33]), quantitative information on the accumulation capacities of lichens compared to other organism (as average values in each sampling site) is very poor [34–36]. This might be because of difficulty to collect and evaluate accumulation capacities for multiple species simultaneously and from a single investigation site. Biazrov (1994) showed ca. 1.4 to 5 times larger $^{137}$Cs accumulation capacity of epiphytic lichens than of barks based on differences in activities per unit dry weight (Bq kg$^{-1}$) [34]. However, $^{137}$Cs accumulation capacity evaluated on a dry weight basis depends on the density and the thickness of the bark sample [29–30]. Sloof and Wolterbeek (1992) evaluated $^{137}$Cs activity in *Xanthoria parietina* (L.) Th. Fr., a foliose lichen, on both a dry weight basis and a contour surface area (epiphytic area) basis (i.e. $^{137}$Cs inventory, Bq m$^{-2}$) [15]. The latter calculations showed relatively constant values. The $^{137}$Cs inventory is therefore a reasonable proxy for the accumulation capacity of lichens, as radionuclide migration from dry or wet deposition of atmospheric fallout depends on the flux density (i.e. the migration rate per unit area).

Currently there is no information on the $^{137}$Cs accumulation capacity of lichens in comparison to the tissues of other organisms such as tree bark from the same growth situation (i.e. from a similar orientation to the FDNPP and growth height). Accumulation capacitates should be calculated by comparing the $^{137}$Cs inventory between lichens and the adjacent tree bark because of the spatially heterogeneous distribution of $^{137}$Cs activity in lichens on trees [17]. This study analysed lichens and bark samples taken from trees near to the boundary site of the Fukushima Dai-ichi Nuclear Power Station (FDNPS) to quantify $^{137}$Cs inventories and accumulation capacities. The investigation site had high $^{137}$Cs deposition levels and was without

anthropogenic interferences, such as decontamination. In addition, we characterised the radioactivity distribution within the samples.

## Materials and methods

### Study site

The study site was located within 1 km westwards of the FDNPS reactor buildings at an elevation of 38 m (37°25'03.40" N– 141°01'13.44" E). Although the site was within the "difficult-to-return zone", we were specially permitted to enter as a legally entitled public authority. The site is located in Okuma town and on land owned by the Tokyo Electric Power Company. Therefore we sought pre-approval for the investigations and sample collections from the town and company. The $^{137}$Cs deposition level at the site was estimated to be over 3,000 kBq m$^{-2}$ by an unmanned helicopter monitoring on 5 September, 2017(https://ramap.jmc.or.jp/map/eng/). The ambient dose equivalent rate at 1 m above the ground was 13 μSv h$^{-1}$ on 14 July 2017. The mean $^{137}$Cs inventory in top 50 mm of soil from 25 samples (Fig 1) was 5,910 ± 3,020 (SD) kBq m$^{-2}$, see ref. [37] for the measurement protocol. Organic layer (litter) samples were also collected within 20 × 20 cm quadrats before soil sampling. The $^{137}$Cs content in these samples ($n$ = 23) was 76.3 ± 66.1 (SD) kBq m$^{-2}$.

### Lichen and bark sampling

A total of 65 foliose macrolichen samples were collected on three *Zelkova serrata* Makino (Z1–Z3) [39 samples; L1–L17 (on Z1), L1–L14 (on Z2), L1–L8 (on Z3)] and three *Cerasus* sp. (C1–C3) [26 samples; L1–L9 (on C1), L1–L11 (on C2), L1–L6 (on C3)] at a height of between 1 and 2 m above the ground on 22, 23 June and 14 July 2017. The barks adjacent to the lichens were also collected from *Zelkova serrata* [27 pieces; B1–B11 (on Z1), B1–B11 (on Z2), B1–B5 (on Z3)] and *Cerasus* sp. [17 pieces; B1–B7 (on C1), B1–B6 (on C2), B1–B4 (on C3)] (Fig 2). These are common lichen and tree species (deciduous broad-leaved trees) in Japan.

The samples of lichens and barks were collected from the tree trunks using a knife. The thickness of the bark samples were ca. 2 mm. Although we tried carefully to collect bark samples without crustose lichens, small colonies with thin thalli of e.g. *Graphis*, *Lecanora*, *Pertusaria*, and other indistinct species that contained radiocaesium may have been present (see Fig

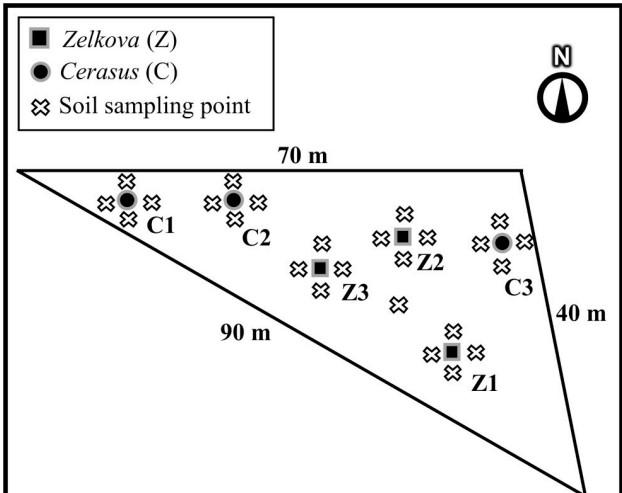

**Fig 1. Sampling situation at the perimeter boundary site of the FDNPS.**

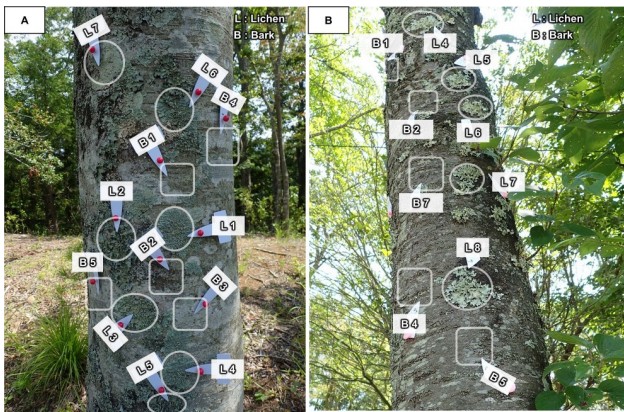

**Fig 2. Sampling points of lichens and barks.** (A) On a tree trunk of *Zelkova serrata*. (B) On a tree trunk of *Cerasus* sp.

2). The samples were obtained from trees that were estimated to be around 30 years old, based on the diameter of the trees and the lichens [38, 39].

Lichen species were identified based on morphological and chemical features. Morphological observations were made using Olympus SZ61 and BX51 microscopes on hand-cut-sections mounted in GAW solution (glycerin: ethanol: water = 1: 1: 1). Chemical substances were detected using thin layer chromatography (TLC) with solvents A, B′ and/ or C [40]. The foliose lichen species identified were as follows: *Canoparmelia aptata* (Kremp.) Elix & Hale, *Dirinaria applanata* (Fée) D.D. Awasthi, *Flavoparmelia caperata* (L.) Hale, *Myelochroa leucotyliza* (Nyl.) Elix & Hale, *Parmelinopsis minarum* (Vain.) Elix & Hale, *Parmotrema austrosinense* (Zahlbr.) Hale, *P. clavuliferum* (Räsänen) Streimann, *P. tinctorum* (Nyl.) Hale, and *Punctelia borreri* (Sm.) Krog.

The lichen samples were cleaned with tweezers to remove bark and any debris. The lichen and bark samples were air-dried at room temperature for three months. Lichen samples for analysis were obtained randomly using a 10 mm diameter hole-punch, and bark samples for analysis were randomly selected and cut into ca. 10 mm square fragments using scissors. These multi fragments (e.g. around 6 pieces per sample) obtained for each individual sample, they were laid on a polypropylene square flat plate (42 mm diameter) and sealed on top of the plate.

## Gamma-ray measurement of $^{134}$Cs and $^{137}$Cs inventories in lichens and barks

In order to evaluate the radioactivity of $^{134}$Cs and $^{137}$Cs (kBq) in the samples, 605 and 662 keV gamma-ray emissions were measured using a high-purity Ge-detector (GMX40P4-76 germanium detector, Seiko EG&G ORTEC) with a relative efficiency of 40% (at 1.33 MeV $^{60}$Co) coupled with a multi-channel analyzer (MCA7600, Seiko EG&G ORTEC). The resolution (FWHM) at 1.33 MeV $^{60}$Co was 2.00 keV. Efficiency calibration of the detector was conducted using a multiple gamma-ray emitting standard source (QCRB21746: mixture NG3). The source contained a known mixture of activities of $^{241}$Am, $^{203}$Hg, $^{139}$Ce, $^{137}$Cs, $^{113}$Sn, $^{109}$Cd, $^{88}$Y, $^{85}$Sr, $^{57}$Co and $^{60}$Co. The source was packed in a plastic disc with an activity area of 42 mm diameter (Eckert & Ziegler Nuclitec GmbH, Braunschweing). The measurement time was 3,600 s. The counting error of the samples were less than 6% for $^{134}$Cs and 2% for $^{137}$Cs in lichen samples and, 17% for $^{134}$Cs and 4% for $^{137}$Cs in bark samples respectively. Since we were mainly evaluating $^{137}$Cs levels in this study, this measurement time was sufficient for the counting error of $^{137}$Cs to be within a few percent. The $^{134}$Cs and $^{137}$Cs radioactivities were

decay corrected to the sampling date. The reported uncertainty values are ± the total uncertainties associated with each sample measurement, i.e. they include both the counting and efficiency calibration errors (S1 Table).

The areas of lichen thallus and bark were measured using a digital microscope with measurement software (VHX-2000, Keyence). The radiocaesium inventory (kBq m$^{-2}$) was calculated by dividing the radioactivity of Cs by the area of lichen thallus and bark, respectively.

### *In-situ* measurement of activity per unit area in lichens and barks

Surface contamination levels of radiocaesium on lichens and barks were measured based on the mainly β-emissions using a Geiger-Müller (GM) survey meter (TGS-146B, Hitachi-Aloka Medical) *in situ* before sampling. Because current radiation was emitted from only radiocaesium in the study area, and no gamma-ray radionuclides contributing to the GM measurement detected significantly except for radiocaesium. Thus, we treated the GM measurement values as entirely originating from radiocaesium since other gamma nuclides were considered to be extremely small if any they were presented. The activities per unit area were evaluated based on JIS standard (JIS Z 4504:2008) [41], which is the Japanese equivalent of ISO standard (ISO 7503–1:1988), as below:

$$A_s = \frac{n - nb}{\varepsilon_i \times W \times \varepsilon_s}$$

$A_s$ = activity per unit area (Bq cm$^{-2}$),
$n$ = counting rate on sample (s$^{-1}$), $n_b$ = counting rate of background (s$^{-1}$),
$\varepsilon_i$ = instrument efficiency, $\varepsilon_s$ = source efficiency,
$W$ = effective area of window (cm$^2$)

Here, since the maximum *β*-emission energy of radiocaesium (both $^{134}$Cs and $^{137}$Cs) are over 400 keV, the $\varepsilon_s$ value was 0.5 [41]. The value of $\varepsilon_i$ was 0.475, and $W$ was 19.6 cm$^2$ for the survey meter used. Each counting rate was measured for 30 seconds using a 10 seconds time constant *in-situ*. The background levels were also measured near the surface of each tree at 1 m height. Each measurement data was shown in S2 Table.

### Autoradiography

Prior to obtaining gamma–ray radioactivity measurements, we visualized the radiocaesium distribution in both lichens and barks by autoradiography using a BAS-IP imaging plate with 25 μm pixels (SR 2040, Fuji Film, Tokyo, Japan). After these samples were attached to the IP for 60 minutes, the autoradiograph images were scanned using a laser scanner (Typhoon FLA7000, GE Healthcare, Little Chalfont, UK). Photo-stimulated luminescence (PSL) signals were also measured using the scanner. The obtained images were analysed using an imaging software (ImageJ 1.52a; free software, https://imagej.nih.gov/ij/).

### Electron microscopic analysis

Lichen and bark samples were cut into small fragments (< 5 mm) using a razor blade. The fragments were mounted onto an aluminum sample table (12.5 mm φ × 10 mm height, JEOL) by carbon tape. To ensure the conductivity of the samples, ca. 12 nm thick osmium was coated onto the samples using an osmium vapor deposition equipment which is suitable for deposition on uneven surfaces (HPC-20, Vacuum Device). We used a field emission electron probe micro analyzer (FE-EPMA) (JXA-8530F, JEOL) to examine the sample surface with an accelerating voltage of 15 kV. A back-scattered electron detector was used to find high atomic

number elements such as metal elements, since the backscattered electron current increases with atomic number. The elemental compositions of particles were measured using energy dispersive X-ray spectroscopy analysis for qualitative spot and map analysis in FE-EPMA.

## Statistical analysis

To investigate the difference in $^{137}$Cs activity concentrations between lichens and barks, and the inventory ratio of $^{137}$Cs in lichens to barks, we checked the normality of the collected data using chi-square ($\chi^2$) tests for goodness of fit ($P < 0.05$). The data were not normality distributed, so we used nonparametric statistical analyses, Mann–Whitney's $U$ test. Normality was also checked for the relationship between radiocaesium inventories measured using gamma spectroscopy and GM measurements. Because these measurement data were also not normally distributed, we tested their relationships by nonparametric correlation analyses (Spearman).

## Results and discussion

All results for the radiocaesium contents in lichens and barks are summarised in Table 1. Both $^{134}$Cs and $^{137}$Cs were measured in all samples. The isotopic ratios ($^{134}$Cs/$^{137}$Cs) decay-corrected to 11 March 2011 ranged from 0.9 to 1.2 which was consistent with the ratio of radiocaesium released from the FDNPP accident (S1 Table) [11, 20, 21].

**Table 1. Radiocaesium activity concentrations and inventories in lichen thalli and tree bark in the site boundary of the FDNPS.**

| Species | Date | n | Activity concentration by gamma spectroscopy (kBq kg⁻¹) | | | | | | Inventories by gamma spectroscopy (kBq m⁻²) | | | | | | n | Inventories by GM measurement (kBq m⁻²) | | |
|---|---|---|---|---|---|---|---|---|---|---|---|---|---|---|---|---|---|---|
| | | | Cs-134 | | | Cs-137 | | | Cs-134 | | | Cs-137 | | | | Cs-134 + Cs-137 | | |
| | | | Mean | ± | SD | Mean | ± | SD | Mean | ± | SD | Mean | ± | SD | | Mean | ± | SD |
| **Soil** | 14/7/2017 | | | | | | | | | | | | | | | | | |
| Organic layer | | 23 | 33.3 | ± | 18.5 | 244 | ± | 137 | 10.4 | ± | 8.9 | 76.3 | ± | 66.1 | | – | | |
| Surface soil 0-5 cm | | 25 | 20.2 | ± | 11.3 | 152 | ± | 85 | 784 | ± | 401 | 5910 | ± | 3020 | | – | | |
| **Lichens (on *Zelkova serrata*)** | 22-23/6/2017 | | | | | | | | | | | | | | | | | |
| *Canoparmelia aptata* | | 1 | 253 | | – | 1710 | | – | 45.2 | | – | 306 | | – | 1 | 187 | | – |
| *Dirinaria applanata* | | 3 | 493 | ± | 171 | 3360 | ± | 1210 | 77.4 | ± | 26.2 | 528 | ± | 184 | 3 | 441 | ± | 66 |
| *Myelochroa leucotyliza* | | 2 | 671 | ± | 435 | 4630 | ± | 2930 | 132 | ± | 84 | 912 | ± | 565 | 1 | 769 | | – |
| *Parmelinopsis minarum* | | 1 | 319 | | – | 2210 | | – | 76.4 | | – | 529 | | – | 1 | 291 | | – |
| *Parmotrema austrosinense* | | 1 | 148 | | – | 1030 | | – | 28.7 | | – | 200 | | – | 1 | 266 | | – |
| *Parmotrema clavuliferum* | | 12 | 357 | ± | 170 | 2430 | ± | 1160 | 69.3 | ± | 26.4 | 472 | ± | 178 | 12 | 381 | ± | 106 |
| *Punctelia borreri* | | 19 | 327 | ± | 92 | 2230 | ± | 620 | 70.8 | ± | 24.9 | 483 | ± | 166 | 19 | 354 | ± | 109 |
| All | | 39 | 360 | ± | 166 | 2460 | ± | 1130 | 72.4 | ± | 31.3 | 494 | ± | 213 | 38 | 372 | ± | 125 |
| **Barks (substrate)** | 22-23/6/2017 | 27 | 9.64 | ± | 8.39 | 67.7 | ± | 58.6 | 14.4 | ± | 12.9 | 101 | ± | 90 | 21 | 103 | ± | 87 |
| **Lichens (on *Cerasus* sp.)** | 14/7/2017 | | | | | | | | | | | | | | | | | |
| *Dirinaria applanata* | | 3 | 221 | ± | 68 | 1590 | ± | 480 | 39.7 | ± | 11.8 | 285 | ± | 82 | 3 | 200 | ± | 88 |
| *Flavoparmelia caperata* | | 1 | 40.6 | | – | 307 | | – | 7.05 | | – | 53.3 | | – | 1 | 102 | | – |
| *Parmelinopsis minarum* | | 1 | 281 | | – | 1930 | | – | 67.4 | | – | 463 | | – | 1 | 219 | | – |
| *Parmotrema austrosinense* | | 6 | 172 | ± | 72 | 1190 | ± | 510 | 30.2 | ± | 15.3 | 209 | ± | 107 | 6 | 189 | ± | 75 |
| *Parmotrema clavuliferum* | | 8 | 128 | ± | 48 | 916 | ± | 317 | 25.3 | ± | 14.5 | 181 | ± | 98 | 8 | 188 | ± | 116 |
| *Parmotrema tinctorum* | | 3 | 107 | ± | 50 | 747 | ± | 332 | 24.5 | ± | 13.5 | 170 | ± | 90 | 3 | 185 | ± | 89 |
| *Punctelia borreri* | | 4 | 230 | ± | 96 | 1610 | ± | 640 | 36.8 | ± | 13.9 | 258 | ± | 93 | 4 | 305 | ± | 29 |
| All | | 26 | 165 | ± | 80 | 1160 | ± | 550 | 30.7 | ± | 16.1 | 216 | ± | 111 | 26 | 205 | ± | 92 |
| **Barks (substrate)** | 14/7/2017 | 17 | 8.29 | ± | 5.47 | 59.2 | ± | 36.5 | 8.01 | ± | 6.79 | 56.2 | ± | 43.3 | 14 | 38.8 | ± | 29.7 |

## Radiocaesium activity concentrations in lichens and barks

Here we focused on $^{137}$Cs, because $^{134}$Cs has a short half-life (2 years) so its activity in all samples had decreased by a factor of around ten over the 6 years following the 2011 FDNPP accident. The mean $^{137}$Cs activity concentrations of lichens were 2,460 ± 1,130 (SD) kBq kg$^{-1}$ [coefficient of variation (CV) = 46%] for *Zelkova serrata* and 1,160 ± 550 (SD) kBq kg$^{-1}$ (CV = 48%) for *Cerasus* sp. In contrast, the mean values in barks were 67.7 ± 58.6 (SD) kBq kg$^{-1}$ and (CV = 87%) for *Zelkova serrata*, and 59.2 ± 36.5 (SD) kBq kg$^{-1}$ (CV = 62%) for *Cerasus* sp., respectively. Lichens showed significantly higher radiocaesium activity concentrations than those of barks (Mann–Whitney *U* test, $P < 0.01$).

In this study, the mean $^{137}$Cs activity concentration in lichens collected six years after the FDNPP accident was approximately 20 to 36 times higher than that of their substrate barks in bulk (Fig 3). Biazrov (1994) showed that the $^{137}$Cs activity concentrations in lichens (*Hypogymnia physodes* and *Cladina mitis*) were 2 to 5 times higher than those of barks (*Pinus silvestris*) at 1.5 km away from the CNPP site 2.7 years after the accident [34]. Furthermore, Sawidis et al., (2009, 2010) demonstrated that the $^{137}$Cs activity concentrations in lichens (total 8–10 species of epiphytic, epilithic and epigeic lichens) were 0.68 to 6.7 times higher than tree barks in the same biota (*Fagus sylvatica*, *P. halepensis* and *Quercus coccifera*) at 2 sampling sites ca. 1,300 km away from the CNPP in Greece 20 years after the accident [36, 42]. In the case of this study for Fukushima Prefecture, the difference of the radiocaesium activity concentrations between lichen and bark was much larger.

The biological half-lives of $^{137}$Cs in lichens are mostly longer (4.9 to 7.9 y in lichens, 0.9 y in mosses), and their uptake rate of $^{137}$Cs (3,300–7,000 Bq kg$^{-1}$ y$^{-1}$ in lichens, 2,000 Bq kg$^{-1}$ y$^{-1}$ in mosses) is higher than other biomonitors, e.g. mosses, according to earlier studies [19, 43, 44]. Based on long term (e.g. longer than 5 years) monitoring studies, $^{137}$Cs effective half-lives of foliose lichens were summarised as 2.6 to 12.9 years by Ramzaev et al. (2016) [27]. The effective half-lives of $^{137}$Cs in barks were estimated as 2.1 to 8.6 years for deciduous broad-leaves trees [*Q. serrata* (Japanese konara oak), *P. densiflora* (red pine) along with some broad-leaved tree species], 1.5 years for mixed broad-leaved stands [*Q. serrata*, *Viburnum furcatum* (forked viburnum) and *P. densiflora*] within the Fukushima Prefecture [45, 46]. Thus, as the first reason, the large difference of $^{137}$Cs activity concentrations between lichens and barks is explained by the difference in retention times of $^{137}$Cs.

While, the inclusion of inner bark with low $^{137}$Cs activity concentrations may have led to an underestimation of the $^{137}$Cs activity concentration in the bark samples. The bark thickness and the shape might be another important factor determining the radionuclide activity concentration in barks [29, 42]. $^{137}$Cs was previously suggested to be absorbed and retained on bark surfaces due to the dead tissues in the outer bark [29]. In case of the FDNPP accident studies, radiocaesium was also distributed on the outer surface of barks [*Castanea crenata* (chestnut tree) and *Cryptomeria japonica* (Japanese cedar)], sampled within the Fukushima Prefecture [47, 48]. In addition, most of the radiocaesium in barks was found in the outer bark of cherry trees (30 years old) [48]. Therefore we collected the outer surface of bark samples with ca. 2 mm thickness, assuming that some inner bark would be included so that outer bark could be sampled. The lichens have a thickness of several hundred micro meters, so the $^{137}$Cs activity concentrations may be artificially lower for barks than for lichens as the former is affected by the mass of the inner parts of the bark. This is the second reason why we found that $^{137}$Cs activity concentrations in barks were lower than lichens. It is clear, therefore, that the radiocaesium inventory, i.e. the $^{137}$Cs activity per unit area, is a more appropriate measure to compare the radiocaesium retention capacities of lichen and bark.

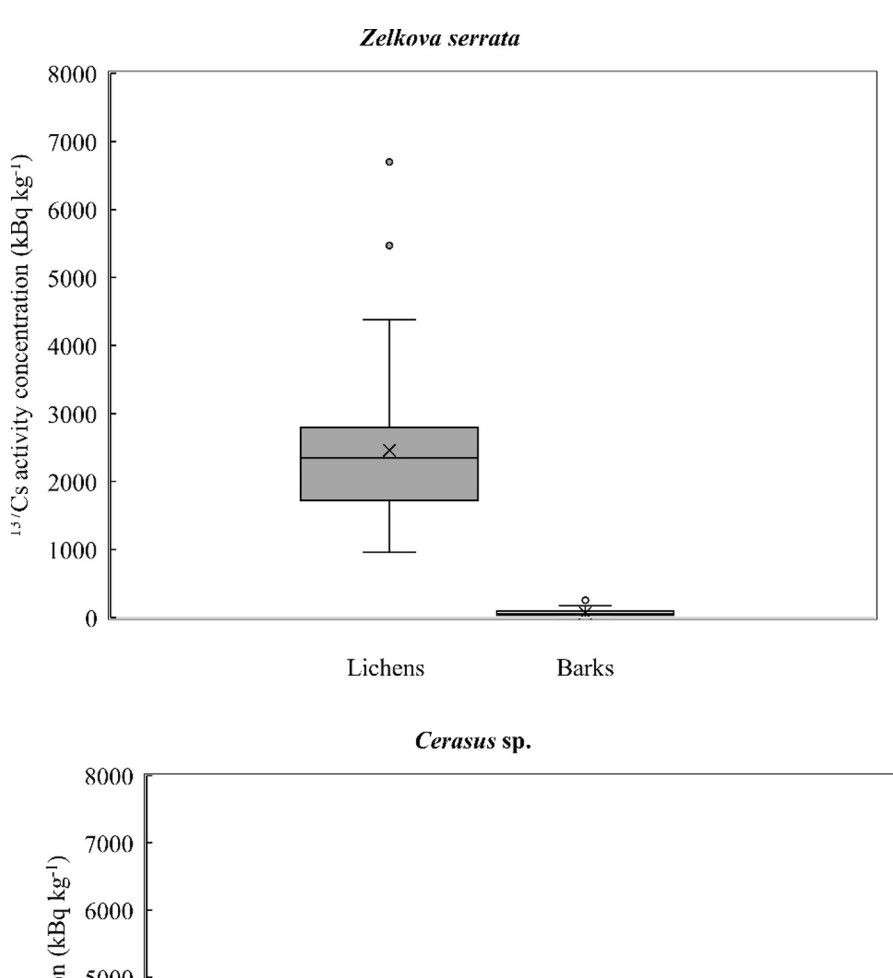

**Fig 3. Comparison of $^{137}$Cs activity concentrations of lichens and barks.** (A) *Zelkova serrata*. (B) *Cerasus* sp.

## Comparison of radiocaesium inventories between lichens and barks

We show the radiocaesium inventories (kBq m$^{-2}$) in Table 1, and compare the radiocaesium inventories ($^{134}$Cs and $^{137}$Cs) determined by gamma spectroscopy and GM measurement of all lichen and bark samples in Fig 4. There was good agreement between the inventories determined by each method. The Spearman's rank correlation coefficient was calculated as 0.92 ($P < 0.01$).

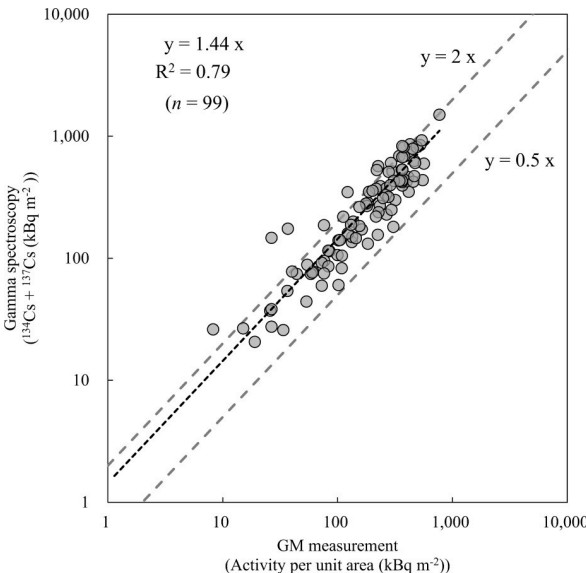

**Fig 4. The relationships between the radiocaesium inventory by gamma spectroscopy and the inventory by GM measurement.**

This study investigated the difference of $^{137}$Cs-inventories between lichens and adjacent barks samples from identical environmental conditions. The ratio of inventories on several lichen sample and adjacent bark were calculated (e.g. $^{137}$Cs-inventrory ratio of lichen Z1–L1 to barks = (Z1–L1) / (mean value of Z1–B1, B2, B3 and B4)), as summarised in Table 2. The mean inventory ratios were 7.9 for *Zelkova serrata* and 3.8 for *Cerasus* sp. This showed that lichens can retain significantly more radiocaesium than barks in the same environmental conditions (Mann-Whitney *U* test, $P < 0.01$).

### Radiocaesium distributions in lichens and barks

The autoradiograms showed black spots in both lichens and barks (Fig 5). The black spots indicate the presence of radioactive particulate matters. There was strong clustering of the spots in the lichen samples, which was not the case for the bark samples. The clustering of the spots suggests that there are particle aggregates and/or particle dissolution and accumulation has occurred. Some barks of *Zelkova serrata* showed radiocaesium distributed along cracks or streaks, but this was not seen in the *Cerasus* sp. samples.

The upper panels are photographs of the samples. The middle panels are autoradiographic images. The lower panels show the three-dimensional distribution of the intensity signals obtained from the autoradiography.

We observed particulate matters on the upper surface of lichens e.g. *Punctelia borreri* and barks using by FE-EPMA analysis (Figs 6 and 7). The particulates ranged from < 1 to several dozen µm in diameter. Spherical and non-spherical particles, mineral-like particles, were found, containing Si, O, Al mainly, and K, Ca, Mg, Ti, Fe, etc. (Fig 6). Micron-sized particulate matters were also observed on bark surfaces in both species (Fig 7). Their characteristics, elemental composition, particle size and shapes, were similar to those found on lichen surfaces (Fig 6).

There are two possible types of radioactive particle causing the black spots in autoradiogram images. The first are mineral radiocaesium particles, e.g. clay mineral particulates with or without organic matters, and weathered biotite [49]. Some of the particles we found on lichen and bark surfaces were similar in elemental composition and size characteristics to such

**Table 2. The inventory ratio of $^{137}$Cs in lichen to in bark.**

| Substrate | Lichen species [a] | n | $^{137}$Cs inventory in lichen / $^{137}$Cs inventory in bark | | | |
|---|---|---|---|---|---|---|
| | | | Mean | SD | Range | CV (%) |
| *Zelkova serrata* | | | | | | |
| | CA | 1 | 8.2 | – | – | – |
| | DA | 3 | 5.2 | 3.2 | 2.0–8.3 | 61 |
| | ML | 2 | 3.6 | – | 3.5–3.8 | – |
| | PnM | 1 | 38.2 | – | – | – |
| | PA | 1 | 3.0 | – | – | – |
| | PC | 12 | 8.6 | 7.1 | 1.3–23.0 | 83 |
| | PuB | 19 | 6.9 | 3.5 | 1.4–15.4 | 51 |
| | **ALL** | 39 | 7.9 | 6.9 | 1.3–38.2 | 88 |
| *Cerasus* sp. | | | | | | |
| | DA | 3 | 5.8 | 1.8 | 3.9–7.3 | 30 |
| | FC | 1 | 1.9 | – | – | – |
| | PnM | 1 | 6.8 | – | – | – |
| | PA | 6 | 3.7 | 2.6 | 1.3–8.1 | 71 |
| | PC | 8 | 3.4 | 2.0 | 1.2–6.4 | 57 |
| | PT | 3 | 4.0 | 3.5 | 1.8–8.1 | 87 |
| | PuB | 4 | 2.8 | 1.0 | 1.8–4.0 | 34 |
| | **ALL** | 26 | 3.8 | 2.2 | 1.2–8.1 | 59 |

[a] CA, *Canoparmelia aptata*; DA, *Dirinaria applanata*; FC, *Flavoparmelia caperata*; ML, *Myelochroa leucotyliza*; PnM, *Parmelinopsis minarum*; PA, *Parmotrema austrosinense*; PC, *Parmotrema claviliferum*; PT, *Parmotrema tinctorum*; PuB, *Punctelia borreri*.

particles. The second are insoluble radiocaesium containing particles, called Cs micro particles (CsMPs), which were emitted from the FDNPS as a result of the accident [50]. Such CsMPs were also found in parmelioid lichens collected nearby the FDNPS [51]. Both types of particle, mineral particles and CsMPs, show as strong black spots in autoradiographs [52]. Therefore, black spots on autoradiograms for lichens and barks indicate the presence of mineral particles containing radiocaesium and/or CsMPs.

Since CsMPs were formed at the time of the FDNPS reactor accidents in March 2011, foliose lichens thus have the ability to retain the CsMPs for at least six years. Based on earlier studies of lichens trapping airborne particles, it is thought that the particles initially accumulate on the thallus surface and trapped between areoles, then accumulate within the medulla where considerable extracellular space exists between loosely interwoven hyphae (e.g. estimated as 18% for *Xanthoria parietina*) [53–56]. In addition, particles integrated in the medullary hyphae were hardly removed after washing, and often resemble dust particles deposited on the substrate [57]. Our FE-EPMA observation showed few micron-sized particles that were caught in parts of the pores of epicortex on the thallus (Fig 6). The mucilageous epicortex covers the thallus surface with a thickness of 0.6–1 μm in some species of Parmeliaceae [58, 59], and may also contribute to particle trapping capacity. Thus, we suggested that particles deposited in the initial fallout were captured and retained on the thallus surface and interspace of the medullary hyphae for a long period owing to adhesion with the mucilage and integration with hyphae. Insoluble particles were trapped in the lichen tissues are expected to maintain their physicochemical properties even after 6 years of the accident.

Autoradiograms for the barks of *Zelkova serrata* showed a spotty distribution along the cracks (Fig 5), and electron micrographic images showed particles on the surface and in the cracks (Fig 7). Similar particles were found on the surface and in the cracks of *Cerasus* sp.

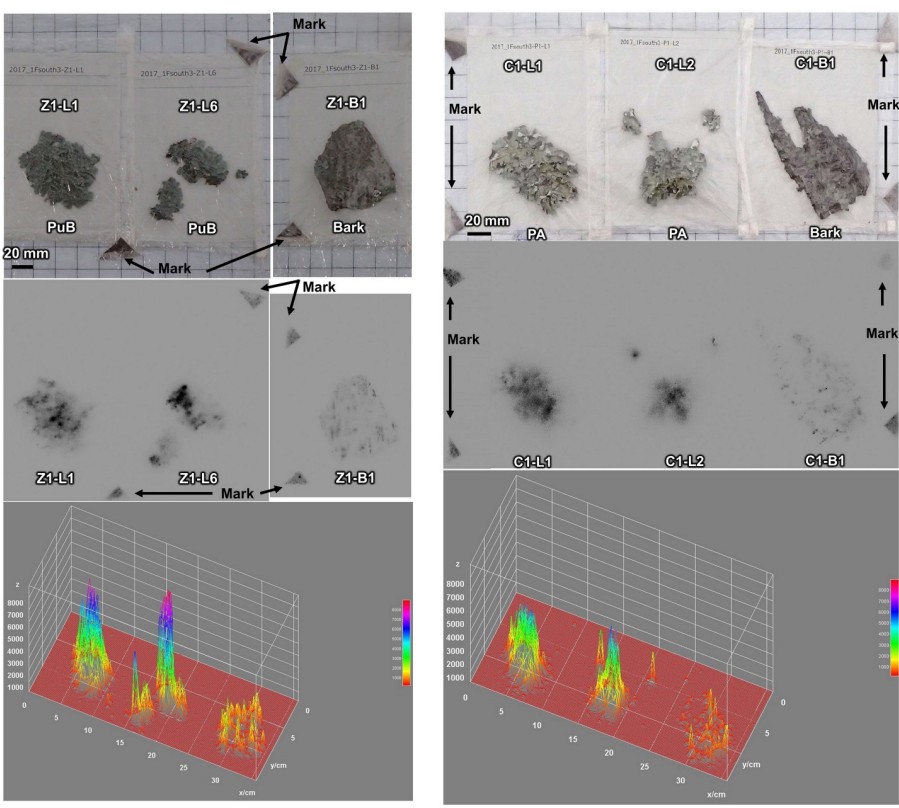

**Fig 5. Autoradiograph images of lichen and bark samples.** (A) *Zelkova serrata*. (B) *Cerasus* sp. PuB, *Punctelia borreri*; PA, *Parmotrema austrosinense*.

although the surface shapes were slightly different from *Zelkova serrata* (Fig 7). This suggests that the particles are physically trapped. Although the total amount and size distribution of the particles in the samples cannot be assessed here, we found that at least particles with similar size and elemental composition were present in both lichens and barks.

We found relatively vague and broad distributions of black spots in lichens and barks (Fig 5) along with strongly clustered areas of spots. Both the intensities and numbers of clusters of spots were higher for lichens than barks. The difference in the intensities between lichens and

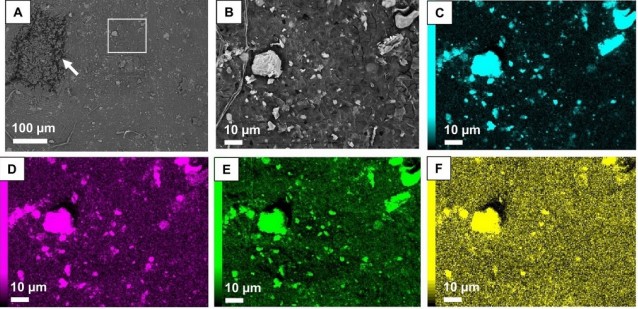

**Fig 6. Back-scattered images (BSI) and the result of map analysis of the upper surface of lichens, *Punctelia borreri* on *Zelkova serrata*.** (A) BSI with a magnification of 200. The square indicates the observation and analysis areas from (B) to (F). The arrow shows pseudocyphellae. (B) to (F) show images with a magnification of 1,000. (B) shows BSI and, each image of (C) to (F) shows the distribution of Si, Al, O, and Mg, respectively.

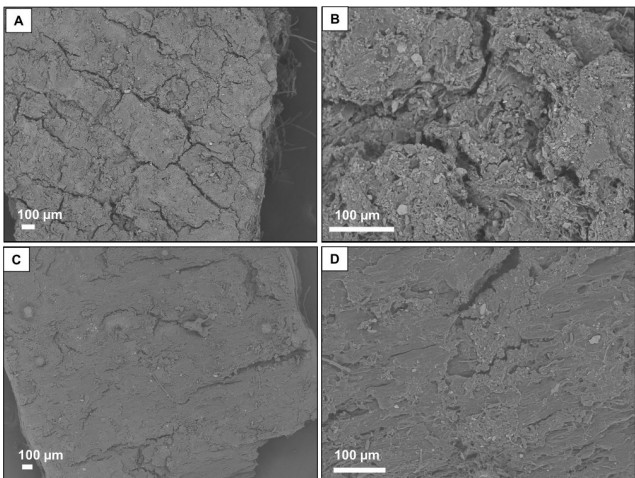

**Fig 7. Back-scattered images of outer bark surface using by FE-EPMA.** (A) (B) *Zelkova serrata* (Z1–B1). (C) (D) *Cerasus* sp. (C1–B1).

barks at the same spot with the same environmental conditions indicates that the lichens have a radiocaesium retention capacity.

We hypothesize two mechanisms to explain the lower retention of Cs by bark. Ion exchange might cause Cs absorption to carboxylic groups, pectin, and phenolic acids of bark tissues, as per the mechanism known for metal ions [60]. This would then allow Cs to be washed away by rain and/or stemflow. A second mechanism may be Cs migration from bark into the tree. Radiocaesium that permeated into inner bark would be rapidly transported radially by ray parenchyma, followed by axial transportation by pith and axial parenchyma [61, 62].

There was more Cs in the central part of lichen thallus than near the edges of the thallus (Fig 5). Since foliose lichens generally grow radially, in which the marginal parts growing faster but the central parts decaying and regeneration [63], such distribution implies a mechanism might keep Cs in the central part of thallus. Lichens are known to accumulate metals absorbing from surrounding environments, e.g. extracellular uptake via ion exchange and intracellular accumulation besides metal-rich particulates entrapment [55, 57, 64]. Our findings suggest that Cs is not readily transferred to new tissues, and that Cs is stored more strongly than ion exchange in the barks. This means that Cs is possibility present in an immobilisation form due to, for instance, transportation intracellularly by chelation and sequestration [65], and formation of metal compounds by lichen secondary metabolites [65–67].

These findings are expected to explain the difference between the radiocaesium uptake and retention process of both particulate and ionic states in lichens and barks. This is due to the different ways nutrients are obtained by lichens and bark (tree), i.e., from the atmosphere in lichens versus through the root system in trees (the role of barks is more to offer physical protection). This would be the reason why lichens retain $^{137}$Cs more efficiency than barks. In this study, we estimated radiocaesium accumulation capacities of lichens and their substrate, bark, under the same growth conditions. We expect to be able to use both as complementary biomonitoring materials in the future.

## Conclusion

In this study, we compared radiocaesium accumulation capacities of lichens and barks quantitatively under the same growth conditions for the first time. We demonstrated epiphytic

foliose lichens had higher $^{137}$Cs activities per unit area than barks of *Zelkova serrata* and *Cerasus* sp. that grow adjacently. We also showed the $^{137}$Cs accumulation characteristics were different between lichens and barks by FE-EPMA and autoradiograph analyses. Both lichens and barks could trap radioactive particles with similar physical characteristics; considered to be around micron sized particles both mineral radiocaesium particles and Cs micro particles (CsMPs). We suggest that lichens are more capable of retaining $^{137}$Cs in immobilised chemical forms within the intracellular structures and their surroundings, whereas the $^{137}$Cs on bark surfaces is reduced by ion exchange and internal migration.

## Supporting information

**S1 Table. Radiocaesium activity concentrations and inventories of all samples by gamma spectroscopy.** Each data was decay corrected to the sampling date. The uncertainty values were included both counting errors and efficiency calibration errors. The $^{134}$Cs/$^{137}$Cs ratios were obtained by decay-correcting the date to 11 March 2011.
(PDF)

**S2 Table. GM counting values and inventories of lichen and bark samples.**
(PDF)

## Acknowledgments

We would like to express our gratitude to the Okuma Town office and the Tokyo Electric Power Company for the permission of field investigations; to Dr. A. Malins for proofreading our manuscript; to Mr. M. Takahashi, Mr. K. Mitachi and Mr. K. Osotsuka for their kind assistances during field investigations and experiments in our laboratory; and to the Fukushima Radiation Measurement Group in JAEA for their contributions to measurement of radioactivity. We are grateful to Mr. S. Kimura for technical assistance in the autoradiograph analysis, to Dr. Y. Sasaki for helpful advice about tree age and to Mr. F. Kanno for assistance to additional test.

## Author Contributions

**Conceptualization:** Terumi Dohi, Yoshihito Ohmura, Kazuya Yoshimura.

**Formal analysis:** Terumi Dohi.

**Funding acquisition:** Terumi Dohi, Kazuki Iijima.

**Investigation:** Terumi Dohi, Yoshihito Ohmura, Seiichi Kanaizuka, Shigeo Nakama.

**Methodology:** Terumi Dohi, Yoshihito Ohmura.

**Project administration:** Kazuki Iijima.

**Visualization:** Terumi Dohi.

**Writing – original draft:** Terumi Dohi, Yoshihito Ohmura, Kazuya Yoshimura.

**Writing – review & editing:** Terumi Dohi, Yoshihito Ohmura, Takayuki Sasaki, Kenso Fujiwara, Kazuki Iijima.

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
