## [Decision Letter · Decision Letter 0]

12 Mar 2021

PONE-D-21-01658

Radiocaesium accumulation capacity of epiphytic lichens and adjacent barks collected at the perimeter boundary site of the Fukushima Dai-ichi Nuclear Power Station

PLOS ONE

Dear Dr. Dohi,

Thank you for submitting your manuscript to PLOS ONE. After careful consideration, we feel that it has merit but does not fully meet PLOS ONE’s publication criteria as it currently stands. Therefore, we invite you to submit a revised version of the manuscript that addresses the points raised during the review process.

In my opinion, your paper is very well done and is largely in very good shape as is. However, Reviewer #1 has made a few suggestion that may be worth incorporating in a final revision. Please consider these suggestions, if suitable.

We look forward to receiving your revised manuscript.

Kind regards,

Tim A. Mousseau

Academic Editor

PLOS ONE

Journal Requirements:

2. In your Methods section, please provide additional information regarding the permits you obtained to collect samples for the present study. Please ensure you have included the full name of the authority that approved the field site access and, if no permits were required, a brief statement explaining why.

3.We note that Figure(s) 1 in your submission contain map images which may be copyrighted. All PLOS content is published under the Creative Commons Attribution License (CC BY 4.0), which means that the manuscript, images, and Supporting Information files will be freely available online, and any third party is permitted to access, download, copy, distribute, and use these materials in any way, even commercially, with proper attribution. For these reasons, we cannot publish previously copyrighted maps or satellite images created using proprietary data, such as Google software (Google Maps, Street View, and Earth). For more information, see our copyright guidelines: http://journals.plos.org/plosone/s/licenses-and-copyright.

a) You may seek permission from the original copyright holder of Figure(s) 1 to publish the content specifically under the CC BY 4.0 license. 

"nothing"

We note that one or more of the authors are employed by a commercial company: Nuclear Engineering Co., Ltd.

Reviewers' comments:

Reviewer's Responses to Questions

**Comments to the Author**

1. Is the manuscript technically sound, and do the data support the conclusions?

Reviewer #1: Yes

Reviewer #2: Yes

2. Has the statistical analysis been performed appropriately and rigorously? 

Reviewer #1: Yes

Reviewer #2: Yes

3. Have the authors made all data underlying the findings in their manuscript fully available?

Reviewer #1: Yes

Reviewer #2: Yes

4. Is the manuscript presented in an intelligible fashion and written in standard English?

Reviewer #1: Yes

Reviewer #2: Yes

5. Review Comments to the Author

Reviewer #1: This study addresses the radiocaesium content of nine epiphytic foliose lichens and the adjacent barks of Zelkova serrata and Cerasus sp . at the boundary of the Fukushima Dai-ichi Nuclear Power Station six years after the accident. They aimed to compare the radiocesium accumulation capacities of lichen and their substrate tree barks. The study is interesting and has been well-written. However, some remarks should be reconsidered. Some more relevant references should be embedded in the text to ensure a deeper discussion. This publication is acceptable if necessary changes are made. My comments/questions/suggestions are appended below:

Minor and major revisions:

19- “nine epiphytic foliose lichens “ pl change to “nine epiphytic foliose lichens species”

20 Please give the English name of the trees.

23 Please change to “lichens (65 specimens) and barks (XX specimen) under the”

Please explain in Intro that why you used lichen and bark as a bioindicator of radiocesium. What are the advantages of those species to be used as a bioindicator of fall-out radionuclides?

Please explain in Intro that what is radiocesium and give some basic info such as physical half-live, sources and elemental properties. The readers might not know the radiocesium and it should be explained briefly in the Intro. Why did you select radiocesium, not other neutron activation and fission products?

37-39 Please refer more relevant studies such as “Belivermiş, M., Çotuk, Y. (2010). Radioactivity measurements in moss (Hypnum cupressiforme) and lichen (Cladonia rangiformis) samples collected from Marmara region of Turkey. Journal of environmental radioactivity, 101(11), 945-951”.

49 “137Cs accumulation capacity evaluated on a dry weight basis depends on the density and the thickness” please give relevant references such as “The usability of tree barks as long term biomonitors of atmospheric radionuclide deposition. Applied Radiation and Isotopes, 68(12), 2433-2437.

90 What was the thickness of the bark you took? It is very important since the radiocesium activity in the inner and outer part of the bark is different.

90 It would good to provide the age of the trees and lichens (at least estimated ages).

110-111 Was the counting geometry identical to the samples? If yes please specify in the text. If no please judge your method.

112-113- Are those values counting error or uncertainty? If they are counting error why are they so high for such an elevated radionuclide activity? If they are uncertainty, please correct the statement and give the details.

114- How did you confirm the accuracy of the measurements? Did you use any standard reference material?

130-133 How did you ensure that all the radiation has been emitted by 134Cs and 137Cs and all are beta. There should be other fission and activation products (at lesser content than radiocesium). GM counter detects all kind of radiation, particularly beta and gamma. It does not allow to differentiate the radionuclide and radiation type. Please provide the necessary details.

160- “beta-ray” please correct that phrase.

161- which non-parametric test you used? Please be clear.

Table 1 “beta-ray” please correct that phrase.

Table 1 I think uncertainty values of the activity concentration should be provided. This table can be kept but all measurement result should be provided (maybe in Supp data) with their uncertainties.

171-176 It would be better to present the lichen species in the MM section.

192- “Our bark samples with ca. 2 mm thick, while the lichens have a thickness of several hundred micro meters.”

The thickness and the shape of the bark is the key factor determining the fall-out radionuclide concentrations in the barks. The outer layer of the bark is composed of dead cells in which radiocesium incorporate but was not eliminated. The thicker the dead layer of the bark the higher retention capacity of atmospheric pollutants. Please see

“The usability of tree barks as long term biomonitors of atmospheric radionuclide deposition. Applied Radiation and Isotopes, 68(12), 2433-2437.” and discuss deeply.

196- The biological half-life of radiocesium is generally comparatively long compared to other bioindicators. Please refer some pieces of literature which has been presented the Tb1/2 of Cs in lichen and other bioindicators if possible. For instance:

Savino, F., Pugliese, M., Quarto, M., Adamo, P., Loffredo, F., De Cicco, F., & Roca, V. (2017). Thirty years after Chernobyl: long-term determination of 137Cs effective half-life in the lichen Stereocaulon vesuvianum. Journal of environmental radioactivity, 172, 201-206.

Topcuoğlu, S., Van Dawen, A. M., & Güngör, N. (1995). The natural depuration rate of 137Cs radionuclides in a lichen and moss species. Journal of environmental radioactivity, 29(2), 157-162.

Papastefanou, C., Manolopoulou, M., & Sawidis, T. (1992). Residence time and uptake rates of 137CS in lichens and mosses at temperature latitude (40 N). Environment international, 18(4), 397-401.

299- “Both lichens and barks could trap radioactive particles with similar physical characteristics” Please be more specific. This study addresses only two radionuclides of one element.

256- spell “aerosols”

285-293 Could you distinguish the algae and fungi part of the lichen thallus by using microscopy or visual observation? In that case it would be very interesting to compare the radiocesium activity in fungal part and algae part. I think it could be done by autoradiography.

Fig 3 a and b: spell “concentration”

Fig 4 “beta ray” should be corrected.

Reviewer #2: Very interesting paper, well written. The idea of combining micro imagine techniques together with study of environmental radioactivity is very interesting and provided interesting results.

I suggest "accept as it is " however I would suggest to correct a typo in name Chernobyl in References.

6. PLOS authors have the option to publish the peer review history of their article (what does this mean?). If published, this will include your full peer review and any attached files.

Reviewer #1: No

Reviewer #2: No

---

## [Author Response · Author response to Decision Letter 0]

1 May 2021

Dear Dr. Tim A. Mousseau and Reviewers,

We are very grateful for editor and reviewers comments, which had help us to improve the manuscript. 

Then, we prepared our response to your comments as an excel file list, and we uploaded it with revised manuscript.

In addiotion, 

- We prepared 2 files (S1 Table and S2 Table) as supporting data.

- We modified Table 1 data about GM measurement, because we found some LOQ data. So, Fig.4 was also reviesed again.

Thank you very much for taking your precious time to check and give comments for our manuscript.

Yours sincerely,

Terumi Dohi

---

## [Editor Report · Decision Letter 1]

4 May 2021

Radiocaesium accumulation capacity of epiphytic lichens and adjacent barks collected at the perimeter boundary site of the Fukushima Dai-ichi Nuclear Power Station

PONE-D-21-01658R1

Dear Dr. Dohi,

We’re pleased to inform you that your manuscript has been judged scientifically suitable for publication and will be formally accepted for publication once it meets all outstanding technical requirements. Congratulations!

Kind regards,

Tim A. Mousseau

Academic Editor

PLOS ONE
---

## [Editor Report · Acceptance letter]

7 May 2021

PONE-D-21-01658R1 

Radiocaesium accumulation capacity of epiphytic lichens and adjacent barks collected at the perimeter boundary site of the Fukushima Dai-ichi Nuclear Power Station 

Dear Dr. Dohi:

I'm pleased to inform you that your manuscript has been deemed suitable for publication in PLOS ONE. Congratulations! Your manuscript is now with our production department. 

Kind regards, 

on behalf of

Dr. Tim A. Mousseau 

Academic Editor

PLOS ONE